# Biological iron-sulfur storage in a thioferrate-protein nanoparticle

Brian J. Vaccaro[1,2,*], Sonya M. Clarkson[2,*,†], James F. Holden[2,†], Dong-Woo Lee[2,†], Chang-Hao Wu[2], Farris L. Poole II[2], Julien J.H. Cotelesage[3], Mark J. Hackett[3,†], Sahel Mohebbi[1], Jingchuan Sun[4], Huilin Li[4], Michael K. Johnson[1], Graham N. George[3] & Michael W.W. Adams[2]

Iron–sulfur clusters are ubiquitous in biology and function in electron transfer and catalysis. They are assembled from iron and cysteine sulfur on protein scaffolds. Iron is typically stored as iron oxyhydroxide, ferrihydrite, encapsulated in 12 nm shells of ferritin, which buffers cellular iron availability. Here we have characterized IssA, a protein that stores iron and sulfur as thioferrate, an inorganic anionic polymer previously unknown in biology. IssA forms nanoparticles reaching 300 nm in diameter and is the largest natural metalloprotein complex known. It is a member of a widely distributed protein family that includes nitrogenase maturation factors, NifB and NifX. IssA nanoparticles are visible by electron microscopy as electron-dense bodies in the cytoplasm. Purified nanoparticles appear to be generated from 20 nm units containing ∼6,400 Fe atoms and ∼170 IssA monomers. In support of roles in both iron–sulfur storage and cluster biosynthesis, IssA reconstitutes the [4Fe-4S] cluster in ferredoxin *in vitro*.

[1] Department of Chemistry, University of Georgia, Athens, Georgia 30602, USA. [2] Department of Biochemistry and Molecular Biology, University of Georgia, Athens, Georgia 30602, USA. [3] Department of Geological Sciences and Chemistry, University of Saskatchewan, Saskatoon, Saskatchewan S7N 5C5, Canada. [4] Cryo-EM Structural Biology Laboratory, Center for Epigenetics, Van Andel Research Institute, Grand Rapids, Michigan 49503, USA. * These authors contributed equally to this work. † Present addresses: Conagen Inc., 15 DeAngelo Dr., Bedford, Massachusetts 01730, USA (S.M.C.); Department of Microbiology, University of Massachusetts, Amherst, Massachusetts 01003, USA (J.F.H.); Laboratory of Applied Biochemistry, School of Applied Biosciences, Kyungpook National University, Daegu 41566, Korea (D.-W.L.); Department of Chemistry, Curtin University, Perth, Western Australia 6102, Australia (M.J.H.). Correspondence and requests for materials should be addressed to M.W.W.A. (email: adams@bmb.uga.edu).

Iron is an essential nutrient for almost all known organisms. It functions as a protein cofactor in fundamental pathways including respiration, photosynthesis and the biogeochemical cycling of sulfur and nitrogen. There are two major types of iron-containing protein cofactors, haemes and iron–sulfur clusters. The most common iron–sulfur cluster is the cubane-type [4Fe-4S] cluster, which is involved in electron transfer, catalysis, DNA repair and small molecule sensing[1]. In spite of their high sensitivity to degradation by oxygen and reactive oxygen species, [4Fe-4S] clusters are ubiquitous in biology. More complex iron–sulfur-containing cofactors containing modified [4Fe-4S] clusters catalyse more chemically challenging reactions in nitrogenase, carbon monoxide dehydrogenase, acetyl-coA synthase and hydrogenase[1].

Iron–sulfur cluster biosynthesis is carried out by two main systems in microorganisms. Bacteria (and mitochondria) typically use the ISC system while archaea and some bacteria (and some plastids) use the SUF system[2]. The ISC and SUF systems have similar mechanisms wherein an iron donor and a cysteine desulfurase provide Fe and S to a scaffold protein, and the metastable scaffold-bound cluster is rapidly delivered to a carrier protein[3]. Due to the complex chemistry and energetic cost involved (ATP is used in initiating cluster release or recruiting Fe and cysteine must be regenerated), it is efficient for a cell to repair iron-sulfur clusters that become damaged by reactive oxygen or nitrogen species, and several repair systems have been proposed[4,5]. In particular, many $[4Fe-4S]^{2+}$ clusters will lose Fe reversibly upon oxidation, to form cubane-type $[3Fe-4S]^{+}$ clusters or cysteine persulfide-ligated $[2Fe-2S]^{2+}$ clusters. Repair of such degraded forms is efficient because they can be restored simply by reduction and the addition of ferrous iron without the need for the full biosynthetic machinery.

While the identity or need for a specific Fe donor for iron-sulfur cluster biosynthesis is still under debate[6], Fe import and storage systems allow cells to avoid the toxicity of free ferrous iron in the presence of $O_2$ while maintaining sufficient cellular iron in spite of the insolubility of free ferric iron at neutral pH[7,8]. In many organisms, Fe is stored in ferritin, which assembles into a 24-mer hollow sphere. Ferritin catalyses the assembly (and release) of a ferrihydrite-type ferric oxy-hydroxide (FeOOH) from $Fe^{2+}$ and $O_2$ in the interior of the sphere using catalytic iron sites, but the mechanism of Fe release from the mineral core is still largely unresolved. Although ferritin homologues exist throughout the three domains of life, including anaerobes, an anaerobic oxidant that facilitates oxidation of $Fe^{2+}$ has yet to be identified, and the physiological role of ferritin in many organisms is not clear[8]. The exact composition of the iron oxide component of ferritin also varies among different species, as does its crystallinity[8]. Abiotic ferrihydrite is disordered or nanocrystalline and porous[9]. Thus this material lends itself to a highly dense form of iron storage that maintains accessibility of the iron due to the high specific area ($\sim300\,m^2\,g^{-1}$) and metastability (relative to other minerals)[9].

To investigate the process of iron and sulfur storage and their incorporation into iron-sulfur clusters in an anaerobic micro-organism that cannot use oxygen to oxidize ferrous iron, we examined the archaeon *Pyrococcus furiosus*, which grows optimally near 100 °C in hydrothermal marine vents[10]. This strict anaerobe grows in the presence of elemental sulfur (S⁰) and uses it as an (insoluble) electron acceptor to generate (soluble) hydrogen sulfide[10]. It has been shown that at least some Archaea use extracellular sulfide directly, rather than cysteine, for iron-sulfur cluster biosynthesis[11]. *P. furiosus* does contain a homologue of the SufS cysteine desulfurase[1,2], but its natural hydrothermal vent environment is typically rich in sulfide and this could be directly incorporated into Fe–S clusters, which

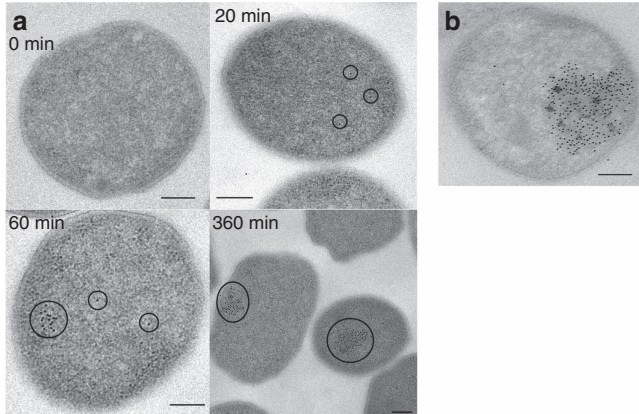

**Figure 1 | Electron micrographs showing IssA assemblies *in vivo*.**
(**a**) The area of IssA localization (immunolabelled with gold nanoparticles) increases over time after addition of S⁰ to a growing culture of *P. furiosus* (scale bars are 200 nm). (**b**) Cells grown using a S⁰ electron acceptor exhibit native electron-dense regions that overlap with immunolabelled IssA (scale bar is 200 nm).

would be much more efficient than cysteine degradation to produce sulfide. In support of this idea, transcriptional analysis comparing *P. furiosus* grown with and without S⁰ revealed upregulation in the expression of numerous genes during hydrogen sulfide production, including those involved in iron and iron-sulfur cluster metabolism[12]. However, the most highly upregulated of these genes (PF2025) encodes a conserved hypothetical protein that at the time was termed sulfur-induced protein A, or SipA (ref. 13). Since expression of *sipA* is only upregulated by sulfide in the presence of sufficient iron[14], this prompted us to investigate whether this protein acts in iron-sulfur cluster metabolism with direct sulfide incorporation. Based on the results presented herein, we rename SipA to the more specific iron–sulfur storage protein A, or IssA.

In this study, we present characterization of IssA. Transmission electron microscopy (TEM) of *P. furiosus* cells expressing IssA shows naturally electron-dense particles that co-locate with IssA immunolabelling. TEM of purified IssA reveals particles up to 300 nm, which appear to be comprised of ~20 nm spheres. X-ray absorption spectroscopy (XAS) strongly supports a thioferrate-type linear $(FeS_2^-)_n$ structure of iron and sulfur, and the EPR of IssA is in accord with this assignment. Finally, we show that IssA is capable of assembling a [4Fe-4S] cluster on *P. furiosus* ferredoxin (Fd), an abundant electron carrier in this organism, in the presence of the small molecule thiol, dithiothreitol (DTT). These properties of IssA together with the conditions under which it is expressed[13,14] lead us to conclude that *P. furiosus* stores excess Fe and S, when they are both highly abundant, in IssA-bound thioferrate, an iron–sulfur structure not previously known in biology. The stored thioferrate can subsequently be mobilized for assembly of [4Fe-4S] clusters, which are widely used in *P. furiosus*. Phylogenetic analyses suggest that homologues of IssA in many archaea, and possibly bacteria as well, may also store iron as thioferrate.

## Results

**Expression of IssA nanoparticles is sulfur-responsive.** When S⁰ was added to a growing *P. furiosus* culture, IssA could be detected in cells by TEM and immuno-gold labelling after 20 min (Fig. 1a). IssA concentration increased over subsequent hours (Supplementary Fig. 1) as well as the area of the cytoplasm occupied by IssA (Fig. 1a), eventually reaching 10–30% of the

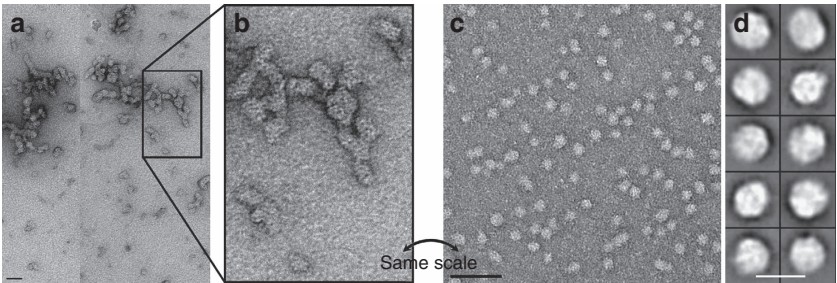

**Figure 2 | Electron micrographs showing IssA assemblies *in vitro*.** (**a**) IssA is purified as nanoparticles of 20–300 nm as shown by negative stain TEM (scale bar is 50 nm). (**b**) Magnified region of **a** to match scale of **c**. (**c**) Treatment of IssA with guanidinium, EDTA and DTT yields a more uniform size distribution of 16–22 nm (scale bar is 50 nm). (**d**) Individual particles from **c** at higher magnification (scale bar is 25 nm).

sectioned cellular area (Fig. 1). IssA is visible not only by immunolabelling but also as electron-dense blotches that co-locate with immunolabelling of IssA both spatially and temporally (Fig. 1). These observations indicate that this protein forms large ($\geq 50$ nm) aggregates *in vivo* after 1 h of IssA expression. Energy dispersive X-ray analysis of the TEM-visualized electron-dense blotches indicated that IssA is associated with iron and sulfur *in vivo* (Supplementary Fig. 2). In addition, proteomic analysis of IssA immunoprecipitated from *P. furiosus* cell extract showed that no other protein is associated with the purified IssA nanoparticle *in vitro* (Supplementary Table 1), suggesting homomeric IssA nanoparticles exist *in vivo*.

**IssA binds iron and sulfur.** IssA was purified from cell extracts of $S^0$-grown *P. furiosus* based on its massive size and high-density relative to other cellular components. Even though the amino acid sequence showed no indication of membrane-association, when cytoplasmic and membrane-associated proteins were separated by ultracentrifugation, IssA was the major protein in the sedimented pellet, as determined by SDS-gel electrophoresis (Supplementary Fig. 3). The membrane-bound proteins in the pellet were dissolved by treatment with detergent (sodium dodecyl sulfate, 1%), however, IssA was not solubilized by this treatment. Density equilibrium centrifugation of the detergent-insoluble material resulted in a black band in which IssA is the only protein present (Supplementary Fig. 3). Metal analysis by inductively coupled plasma mass spectrometry (53 elements) of purified IssA showed that it contains approximately 38 iron atoms and 1 zinc atom per 19 kDa IssA monomer (Supplementary Table 2). Colorimetric assays indicated that IssA contains 38 acid-labile sulfide ions and 17 sulfane sulfur atoms per protein monomer. The interpretation of the sulfur measurements is discussed further below.

**Characterization of IssA nanoparticles.** Sedimentation of IssA after ultracentrifugation was consistent with the observation of large particles via TEM. This was investigated further using size exclusion chromatography. Most ($\sim 90\%$) of the protein was too large to enter the chromatography column and was retained on the pre-column filter (diameter 1 µm). IssA that entered and eluted from the chromatography column did so in a peak at the exclusion limit (40 MDa) or just after the exclusion limit (100 MDa dextran or 400 nm spheres) from Superose 6 and Sephacryl S-1000 SF columns, respectively (Supplementary Fig. 4). Dynamic light scattering analysis was attempted to obtain more precise sizing information. However, reproducible results could not be obtained from IssA. Centrifugation to minimize dynamic light scattering interference by small air bubbles led to sedimentation of IssA, overall signal was low due to

the high extinction coefficient of IssA samples (Gregory L. Hura, personal communication), which appear black, and IssA particles are not uniform in size, as was observed on electron micrographs.

Negative stain TEM of the purified protein showed IssA assemblages with dimensions ranging from 20 to 300 nm (Fig. 2a). Incubation of IssA with 3 M guanidinium chloride, 0.5 mM EDTA and 25 mM DTT, however, yields a more monodisperse sample comprised of roughly spherical particles 16–22 nm in diameter (Fig. 2b). Thus it appears that the 19 kDa IssA monomer forms modular nanostructures based on a $\sim 20$ nm packing unit, containing $\sim 6,400$ Fe atoms and $\sim 170$ IssA monomers. A protein complex of this size of average density would have a mass of $\sim 3.5$ MDa.

**XAS and EPR spectroscopy of IssA-bound iron and sulfide.** XAS was conducted at both Fe K-edge and S K-edge absorption energies to characterize the iron and sulfur bound by IssA. Figure 3a compares the X-ray absorption near-edge spectra of IssA with a number of different iron–sulfur proteins. For both the S and Fe K-edges, the spectra strongly resemble those of the linear [3Fe-4S]$^+$ cluster in the high pH form of the enzyme aconitase[15]. The sulfur K-edge data of IssA show two pronounced peaks, one at low energy at 2467.1 eV, and a broader less-well defined feature at 2470.2 eV. The former, lower energy absorption, is characteristic of sulfide co-ordinated to ferric ions, arising from dipole-allowed transitions of the $1s$ electron to unfilled molecular orbitals involving both sulfur $3p$ and metal $3d$ orbitals, with vacancies due to covalency of the Fe–S bond [16,17]. The second, higher energy transition is attributable to other types of sulfur in the system, such those in the one cysteine and four methionine residues in the IssA monomer (see below). In both S and Fe K-edge cases, the IssA spectra are highly characteristic of a Fe(III) oxidation state[16,18].

The S and Fe K-edge EXAFS oscillations, together with the corresponding EXAFS Fourier transforms, are shown in Fig. 3b, with the best fits. The Fe EXAFS data is dominated by intense backscattering from four Fe–S interactions at 2.24 Å, plus backscattering attributable to two Fe⋯Fe interactions at 2.70 Å. The sulfur K-edge EXAFS, on the other hand, fits to two S–Fe interactions, with no substantial outer shell contributions (Fig. 3b). Experiments at the low X-ray energies of the sulfur K-edge are more challenging than at the iron K-edge, and for this reason the latter data are of substantially better signal to noise ratio. The iron EXAFS also shows a second less intense outer shell Fourier transform peak at 5.4 Å, twice that of the shorter Fe⋯Fe interaction. This fits well to a long-range Fe⋯Fe interaction once multiple scattering interactions for a linear arrangement of iron atoms are included. This 5.4 Å interaction

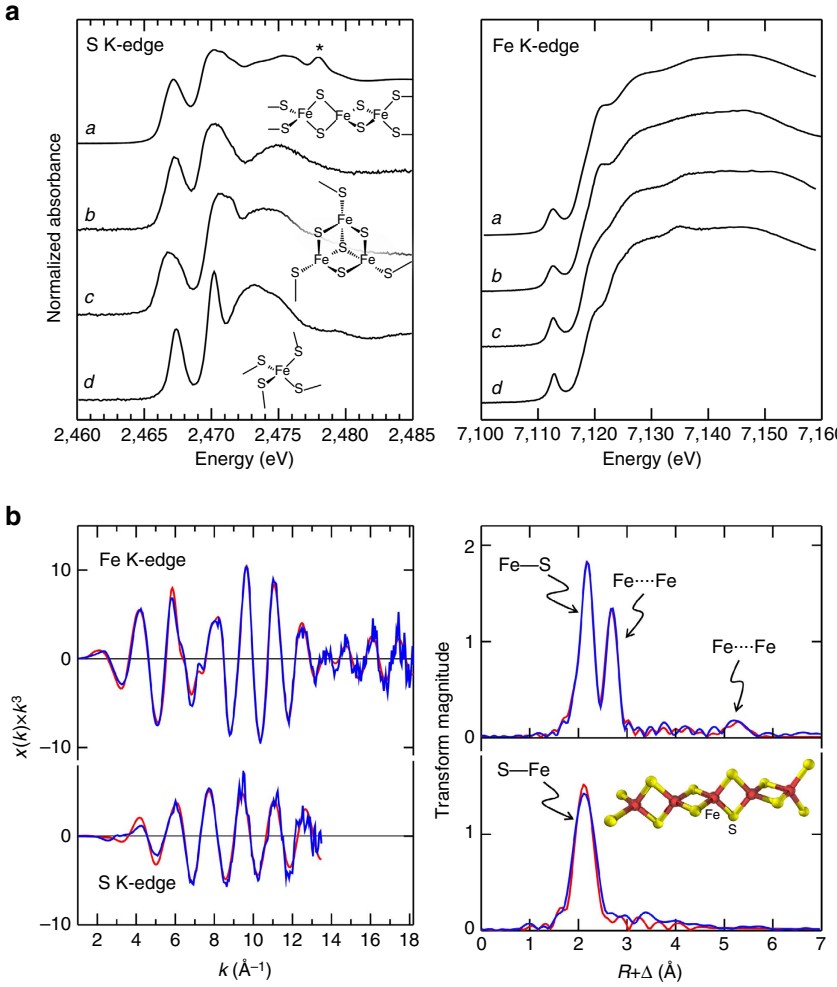

**Figure 3 | IssA X-ray absorption spectra.** (**a**) X-ray absorption near-edge spectra of IssA compared with a number of Fe–S proteins; a: IssA, b: linear 3Fe-4S cluster in human aconitase, c: *P. furiosus* 3Fe-4S ferredoxin and d: *P. furiosus* rubredoxin. The feature marked (*) in the IssA spectrum is due to a trace amount of sulfonate buffer. For both the S and Fe K-edge data, the IssA spectrum most resembles that of the linear 3Fe-4S cluster. (**b**) S and Fe K-edge EXAFS spectra, together with EXAFS Fourier transforms (S–Fe and Fe–S phase-corrected, respectively) showing experimental data (blue lines) together with best fits (red lines), the inset in the Fourier transform figure shows the structure used to compute the multiple scattering EXAFS. Best fits were computed with two S-Fe at 2.239(3) Å, $\sigma^2 = 0.0049(3)$ Å$^2$ and four Fe-S at 2.243(1) Å, $\sigma^2 = 0.0044(1)$ Å$^2$, two Fe $\cdots$ Fe at 2.704(1) Å, $\sigma^2 = 0.0032(1)$ Å$^2$ and two Fe $\cdots$ Fe at 5.408 Å and $\sigma^2 = 0.0064$ Å$^2$.

is close to the limit of the noise, being about three times the transform peak height of the noise as estimated from higher $R$ values, using data to 18 Å$^{-1}$. However, it shows behaviour characteristic of real EXAFS, rather than a noise peak in the Fourier transform, so that the feature persists irrespective of the $k$-ranges, and moreover fits to a very similar Fe $\cdots$ Fe distance with different $k$-ranges. The use of multiple scattering EXAFS reproduces many weaker features in the EXAFS (Fig. 3b) that also appear to be above the noise level (Supplementary Fig. 5). Other weak interactions in the EXAFS data, such as the 3.3 Å feature in the S K-edge EXAFS, do not behave in this manner and these are likely due to noise. Similar long-range Fe $\cdots$ Fe interactions have previously been observed in the Fe K-edge EXAFS of aconitase containing a linear [3Fe-4S]$^+$ cluster[15].

Taken together these XAS data indicate that IssA contains a linear (FeS$_2^-$)$_n$ polymer with two sulfur atoms bridging each pair of Fe(III) ions. Compounds with such a structure are known as thioferrates[19,20]. They have been synthesized intentionally[21] and unintentionally[22] and occur naturally as the mineral erdite[23], however, this is the first time they have been found in a biological system. Thioferrates contain Fe$^{3+}$ and S$^{2-}$ in anionic chains of edge-sharing FeS$_4$ tetrahedra separated by charge-balancing cations (Fig. 4), and the oxidation states and atomic structure are in agreement with our conclusions for IssA based on XAS. In agreement with this hypothesis, the sulfur K near-edge spectrum of CsFeS$_2$ has been reported[17], and shows a distinctive low-energy peak at 2467.0 eV (allowing for the different energy calibration used by Rose *et al.*[17]). Other mineral forms of iron sulfide are inconsistent with the EXAFS because they would show more than two short-range Fe $\cdots$ Fe interactions and would lack the observed long-range 5.4 Å Fe $\cdots$ Fe interactions. Moreover, discrete clusters such as the aconitase linear [3Fe-4S] cluster would require protein-based external thiolate donors that are not present in IssA (it contains only one Cys residue/monomer, see below). We therefore conclude that IssA contains polymeric ferric sulfide with a thioferrate-type (FeS$_2^-$)$_n$ structure. This is also supported by the measured acid-labile iron, sulfide and sulfane sulfur content of IssA (38 Fe, 38 S$^{2-}$ and 17 S$^0$ atoms/monomer). Degradation of synthetic thioferrates in acid (the conditions used for the assays) has been shown to produce Fe$^{2+}$, S$^{2-}$ and S$^0$ in a 2:3:1 ratio[22], which explains the detection of 'S$^0$' in a 1:2 ratio with iron.

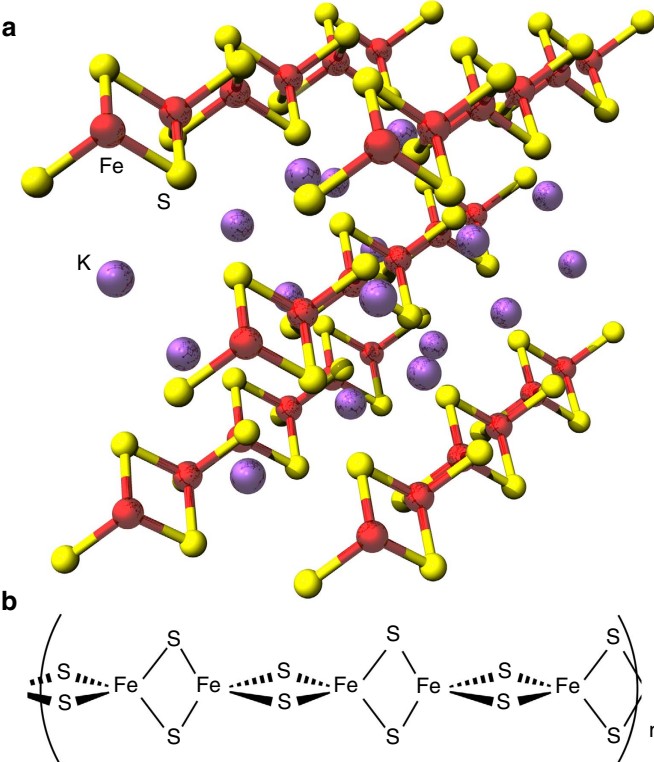

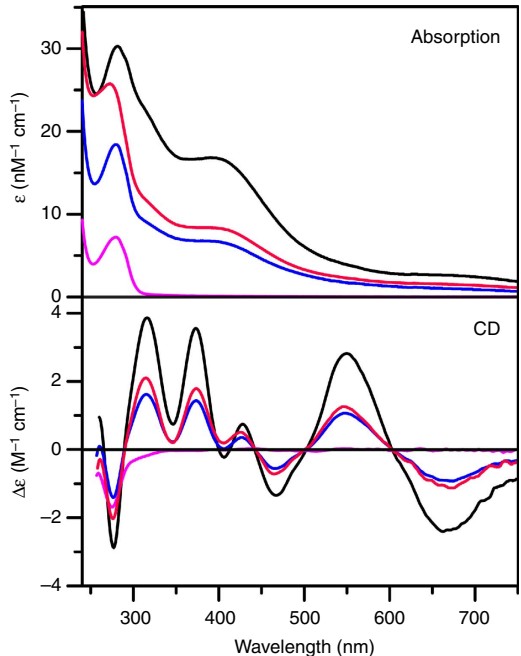

**Figure 5 | IssA-mediated reconstitution of a [4Fe-4S]$^{2+}$ cluster.**
Reference UV-visible and CD spectra for holo- and apo- forms of *P. furiosis* Fd are shown by black and magenta lines, respectively. Repurified Fd products of IssA-mediated reconstitution of apo-Fd in the presence of DTT are shown in red (room temperature for 24 hours) and blue (80 °C for 1 hour).

**Figure 4 | Thioferrate structure. (a)** Inorganic potassium thioferrate (KFeS$_2$) supercell of the crystal structure using three times the crystallographic *c* axis. Potassium is shown as purple spheres, iron as red spheres and sulfur as yellow spheres. Sections of five linear FeS$_2$ chains are visible. **(b)** A five Fe section of a single thioferrate chain, the proposed Fe–S component of IssA.

Since S$^0$ will also be produced under the acidic conditions of the S$^{2-}$ assay, formation of polysulfides further reduces the amount of sulfur available for detection as H$_2$S (ref. 24).

IssA exhibits a very broad, isotropic-type EPR signal spanning ~3,000 Gauss and centred around $g \sim 2.2$. It is only observable above ~60 K and increases in intensity with increasing temperature (Supplementary Fig. 6). At 60 K, the spectra show weak resonances centred around *g* values of 4.3 and 2.0, which increase in intensity with decreasing temperature. The $g = 4.3$ signal is indicative of trace amounts of adventitiously-bound high-spin $(S = 5/2)$ Fe$^{3+}$ or magnetically isolated linear [3Fe-4S]$^+$ clusters $(S = 5/2)^4$. The origin of the weak signal in the $g = 2$ region (<0.01 spin/IssA monomer) is unknown. The anomalous temperature dependence of the very broad isotropic signal is unique among protein-derived EPR signals, but is in agreement with the EPR signal observed for synthetic thioferrates[20,25]. Generally, EPR signals increase in intensity with decreasing temperature, however, the antiferromagnetic coupling between high-spin Fe$^{3+}$ ions in thioferrate polymers results in decreased intensity at temperatures below the Néel temperature[26]. At elevated temperature, thermal disordering of the orientation of electron spins results in increased net magnetization and synthetic thioferrates yield a broad, isotropic EPR signal centred near $g = 2$ (refs 20,25).

***In vitro* IssA-mediated reconstitution of apo-Fd.** To test the hypothesis that IssA functions as a storage protein for Fe and S that can be used for the assembly of iron-sulfur clusters, the ability of IssA to reconstitute the [4Fe-4S] cluster in the apo-form

of *P. furiosus* ferredoxin (Fd) was investigated. *P. furiosus* almost exclusively contains [4Fe-4S] cluster-containing Fe–S proteins, and *P. furiosus* Fd is an abundant protein that is used as electron donor for numerous enzymes. Reconstitution experiments were carried out anaerobically under a variety of conditions. IssA with stoichiometric or a twofold excess of bound Fe and S (as thioferrate) was mixed with apo-Fd at pH 6.8 (the physiological pH for *P. furiosus* growth) and incubated at room temperature for 24 h or at 80 °C for 1 h. Since a [4Fe-4S]$^{2+}$ cluster is more reduced than the all-Fe$^{3+}$ thioferrate iron and sulfide donor, sodium dithionite ($E_m \sim -420$ mV versus NHE), DTT ($E_m \sim -330$ mV versus NHE), or tris(2-carboxyethyl) phosphine (TCEP; $E_m \sim -280$ mV versus NHE) were included in the reaction mixture as a reductant, a disulfide-cleaving reagent (DTT and TCEP) and a dithiol-metal chelating agent (DTT). After centrifugation to remove unreacted IssA, the Fd was purified and the cluster content and integrity were assessed compared with native holo-Fd based on ultraviolet–visible absorption and CD spectra quantified based on protein determinations. No reconstitution occurred in the presence of dithionite or TCEP. However, reconstitution of the [4Fe-4S]$^{2+}$ cluster was observed in the presence of DTT (Fig. 5). Reconstitutions with a twofold excess of IssA-bound Fe and S resulted in 50 ± 5% cluster incorporation after incubation with IssA for 24 h at room temperature. Cluster incorporation was greatly accelerated at more physiologically relevant temperatures, with 45 ± 5% cluster incorporation after incubation with near stoichiometric IssA Fe and S at 80 °C for 1 h. These results provide *in vitro* evidence that the IssA thioferrate core can provide iron and sulfide for [4Fe-4S] cluster assembly in *P. furiosus*. They also suggest that *in vivo* an as yet unknown thiol(s), that is replaced *in vitro* by DTT, plays a role in disassembling the thioferrate polymer into transferrable pieces by

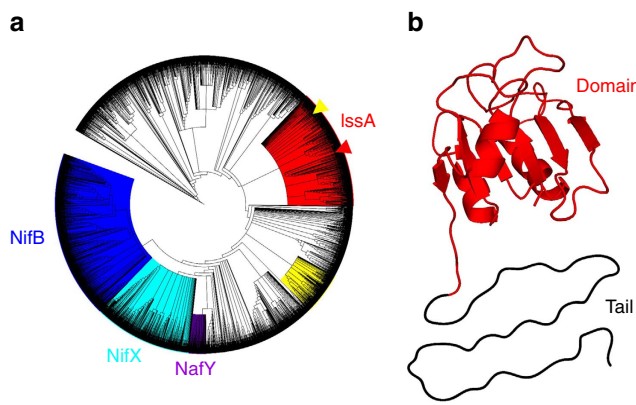

**Figure 6 | IssA bioinformatic analysis.** (**a**) Cladograms of proteins containing the IPR003731 domain. Coloured clades contain members of the NifB (IPR005980, blue), NifX (IPR013480, turquoise) and NafY (IPR031763, purple) InterPro protein families. The proposed IssA clade (red) is based on predicted protein isoelectric point (pI) and occurrence of proteins with a glycine rich region at the C-terminus. *P. furiosus* IssA and the homologue from *Methanothermobacter thermautotrophicus* (MTH1175) are shown by red and yellow arrowheads respectively. (**b**) Modelled structure of the IssA IPR003731 domain based on the NMR structure of MTH1175 (red). The C-terminal region is shown unstructured, as predicted for the apo-form.

chelating Fe, [2Fe-2S]$^{2+}$, or linear [3Fe-4S]$^{1+}$ fragments under reducing conditions that are then assembled into [4Fe-4S]$^{2+}$ clusters in acceptor proteins such as Fd.

**Phylogeny of IssA is distinct from Nif-related families.** IssA from *P. furiosus* is composed of 179 amino acids and has a predicted molecular weight of 19 kDa from its gene sequence (PF2025). The N-terminal 109 residues comprise an IPR003731 InterPro globular domain[27] while the C-terminal residues form a tail region that contains the one cysteine residue in the protein. The globular domain has been identified in over 5,000 proteins widely distributed throughout Archaea and Bacteria. These include many hypothetical proteins as well as members of the NifB, NifX and NafY protein families that function in the maturation of the iron–molybdenum–sulfur (Fe–Mo–S) cluster used by nitrogenase. A cladogram of all proteins containing the IPR003731 domain (Fig. 6a) clearly differentiates a NifX clade (84% bootstrap confidence) and a closely related NafY clade (100% confidence). NifB proteins, which also contain a radical-SAM domain, form a distinct clade (92% confidence) that is the most distantly related of the Nif proteins to the IssA clade (78% confidence), which is based on the *P. furiosus* protein.

NifB, NifX and NafY all bind complex iron-sulfur clusters and transfer them to other proteins. NifB and NifX bind a precursor (NifB-co)[28] in the biosynthesis of the nitrogenase Fe–Mo–S cluster (MoFe$_7$S$_9$C)[29], which is bound by NafY. NifB-co is assembled on the IPR003731 domain of NifB with a proposed composition of Fe$_8$S$_9$C (ref. 30). NifB proteins contain many highly conserved cysteine residues that bind the Fe–S cluster precursors to NifB-co. However, most IPR003731 domain-containing proteins (2,948) contain fewer than the four cysteines required to fully ligate an iron-sulfur cluster (Supplementary Fig. 7). Due to sequence diversity in this large InterPro family, there is little broad amino acid conservation. However, members of the IssA clade contain conserved acidic residues suited to binding the ferric iron necessary for thioferrate synthesis. Thus, we propose that the IPR003731 domain in *P.*

*furiosus* IssA functions as a scaffold for thioferrate assembly and binding, analogous to the role of the homologous NifB domain in NifB-co assembly. Unlike thioferrate, NifB-co likely contains ferrous iron[30,31]. However according to the proposed Fe$_6$S$_9$C composition[30], it is likely an anionic iron–sulfur species like thioferrate (overall charge of $-2$ or $-4$ (refs 30,31)) and may require electrostatic interactions with the protein for stable binding.

**The IssA IPR003731 domain binds iron and sulfur.** A gene encoding the polyhistidine-tagged globular IPR003731 domain of IssA (residues 1–109) was constructed, expressed in *E. coli,* and purified by affinity chromatography (apo-IssA) to investigate whether the domain binds iron and sulfide and how this affects oligomerization. When colourless monomeric apo-IssA was incubated with a 40-fold excess of iron (ferrous ammonium sulfate) and sulfur (sodium sulfide), the black product eluted from a Superose 6 SEC column with molecular weights ranging from ~200 kDa to ~900 kDa with up to 25 Fe per IssA monomer. Hence, the IPR003731 globular domain of IssA clearly binds Fe and S, stimulating oligomerization *in vitro*, although the product is very heterogeneous and it was not characterized further.

**Chemical character and conservation of the IssA C-terminus.** The C-terminal tail (70 of 179 residues) of *P. furiosus* IssA contains a proline-rich region (9 prolines in 23 amino acids; or 11 in 70) followed by a flexible region comprised predominantly of cationic (7 of 70), aromatic (14 of 70) and glycine (15 of 70) residues. Secondary structure prediction[32] indicates that this tail is unstructured. We propose that this cationic region (the predicted tail pI is 10.7) is involved in binding thioferrate formed by the N-terminal domain. Many members of the IssA clade (Fig. 6a) also have a positively charged tail region with a high aromatic and glycine content (Supplementary Fig. 7) although the sequence of the tail in the *P. furiosus* protein is highly conserved only in IssA proteins in species within the Thermococcales. Indeed, some members of the IssA clade have a shortened tail (<70 residues) and some have little or no tail region, which includes the IssA-type protein from *Methanothermobacter thermautotrophicus* that was used to model the structure of the IPR003731 domain of *P. furiosus* IssA (Fig. 6b). In spite of a lack of high sequence conservation in long-tailed IssA members the prevalence of key residues (tryptophan, glycine and arginine) is conserved and a similar repeating pattern is present thereby conserving the chemical character of the C-terminal region of these IssA-like proteins. This is consistent with their proposed role in binding electrostatically to thioferrate, which is negatively charged along its length and does not require ligands with precise positioning. Interestingly, the cladogram reveals another clade of non-themrophilic IssA homologues (termed IssX, Supplementary Fig. 7) that share key features with IssA including a high pI and a glycine rich tail, although none have yet been characterized.

**Discussion**
The results presented herein show that IssA is a novel type of protein with a unique role in iron–sulfur metabolism. As indicated by TEM analysis of natively purified protein, it assembles into massive polymeric nanostructures reaching 300 nm in size. XAS analyses indicate that Fe and S are bound to the protein in thioferrate-type linear chains of ferric sulfide (Figs 3 and 4). While other large monomeric metalloprotein complexes are known (~12 nm diameter ferritin[33] and a 30–35 nm diameter calcium-binding piscine betanodavirus capsid[34]), IssA forms much larger particles and by far the

most iron of any complex known (96,000 in a 300 nm string of 20 nm spheres), though less iron per kDa of protein than ferritin (1.8 Fe/kDa versus 9.6 Fe/kDa, respectively)[8]. Treatment of purified IssA with guanidinium, EDTA and DTT in an aerobic environment leads to homogeneous, spherical particles of ~20 nm diameter (Fig. 2b), which may comprise a basic unit of assembly for larger structures observed in vivo (Fig. 1a,b) and in vitro (Fig. 2a). These treatment conditions are capable of destabilizing protein structure as well as likely degrading thioferrate through chelation of ferric iron by EDTA and removal of sulfide to form DTT persulfide from DTT disulfide. Thus it seems likely that thioferrate is involved in holding these 20 nm units together in the larger nanostructures. A 20 nm sphere would be expected to contain approximately 6,400 Fe atoms and 170 copies of the 19 kDa polypeptide. Ferrihydrite iron, found in ferritin, is accessible to iron removal due to its low crystallinity and high specific area[9]. Similarly, thioferrate iron may be relatively accessible due to its one-dimensional structure, requiring fewer bond scissions for iron removal than three-dimensional minerals.

We have also demonstrated that IssA can provide Fe and S for assembly of $[4Fe-4S]^{2+}$ clusters in apo-Fd from P. furiosus under anaerobic conditions in the presence of DTT (Fig. 5). The mechanism of this process has yet to be determined, but likely involves the ability of DTT (in vitro) or a cellular thiol (in vivo) to bind $Fe^{3+}$ and/or $[2Fe-2S]^{2+}/[3Fe-4S]^{1+}$ thioferrate fragments (Fig. 4). For example, DTT is known to ligate $Fe^{3+}$ and $Fe^{2+}$ (ref. 35) and the non-cysteinyl-ligated sites of protein-bound $[4Fe-4S]^{2+}$ clusters[36]. Moreover, a $[2Fe-2S]^{2+}$ or $[3Fe-4S]^{+}$ fragment of thioferrate could theoretically be obtained from IssA by simple ligand exchange if appropriately activated (for example, by reduction), and both clusters can be readily converted to $[4Fe-4S]^{2+}$ clusters in biological and in synthetic chemistry[37,38]. The conversion of two $[2Fe-2S]^{2+}$ clusters to generate a $[4Fe-4S]^{2+}$ cluster occurs via two-electron reductive coupling[37,38], and this reaction is believed to be involved in de novo cluster assembly in the ISC system[38,39]. In aconitase and in synthetic chemistry, linear $[3Fe-4S]^{1+}$ clusters can be converted to a $[4Fe-4S]^{2+}$ cluster by the addition of $Fe^{2+}$ and one electron[37]. While we have yet to determine the physiological mechanism for Fe–S cluster assembly in P. furiosus, the in vitro cluster assembly results presented here, coupled with the high Fe and S content of IssA and its iron- and sulfide-dependent expression, strongly support a role for IssA in storing Fe and S that can be used for the biosynthesis of Fe–S clusters. Moreover, the in vitro results raise the possibility of spontaneous [4Fe-4S] cluster assembly from $Fe^{3+,2+}$ and $S^{2-}$ on acceptor proteins under Fe and sulfide replete conditions in some strictly anaerobic hyperthermophilic archaea such as P. furiosus. Except for SufCBD and two putative SufS cysteine desulfurases, P. furiosus does not encode any other known Fe–S cluster assembly protein. Moreover, SufC and SufD contain no cysteine residues and the putative SufB scaffold protein has only two cysteines (compared to 13 in E. coli SufB), which are both rigorously conserved in other SufB proteins. Hence, the scaffolding hypothesis that constitutes the current paradigm for Fe–S cluster assembly[39], may not apply in P. furiosus and related organisms.

Because IssA is only produced in P. furiosus when the organism is grown in the presence of abundant iron and sulfide, we propose that the thioferrate structure is synthesized directly from these inorganic precursors. The fact that apo-IssA binds Fe and S from inorganic salts also supports this idea. Interestingly, expression of the genes encoding the two cysteine desulfurase homologues in P. furiosus (PF0164 and PF1066) are strongly down-regulated (5.1- and 3.2-fold) in response to $S^0$, while the sufBD homologues, which are likely to be involved with some aspect of Fe–S cluster trafficking, are strongly upregulated along with issA when $S^0$ is present[12]. Thus, sulfide itself rather than cysteine is likely to provide sulfur for Fe–S cluster assembly under conditions of intracellular sulfide production in P. furiosus. Consequently, Fe–S clusters synthesized from thioferrate may require less energy than canonical ATP-driven scaffold-assembled Fe–S clusters.

Binding of inorganic Fe and S to IPR003731 domain of apo-IssA suggests that native IssA binds anionic thioferrate $(FeS_2^-)_n$ at the IPR003731 domain and that the tail region stabilizes the structure through electrostatic interactions. The 70 amino-acid C-terminal region of IssA contains 7 cationic residues (mostly arginine) that are sufficiently close to each other to preclude a folded structure without a negatively charged counterpart such as thioferrate. In addition, the abundance of glycine residues further indicates a lack of secondary structure in the absence of thioferrate. Hence we propose a model in which the cationic tail may bind the anionic thioferrate chain by wrapping around thioferrate in perhaps a helical arrangement, which is further stabilized by interactions between the tail's aromatic residues. This interaction would confer a defined structure on the otherwise disordered IssA tail. Since apo-IssA is purified as a monomer, but oligomerizes in the Fe–S bound state, we suggest that formation of the observed ~20 nm spherical IssA particles is also dependent on association with thioferrate. According to the estimated Fe-protein ratio, additional cations are needed to completely balance the negative charge on thioferrate, and we expect these are provided by loosely bound cations as well as the single zinc ion per protein (Supplementary Table 1), which may also play a structural role in nanoparticle formation.

## Methods

**Thin section electron microscopy.** P. furiosus was grown as described[12]. Cells were fixed with 2% paraformaldehyde and 1% glutaraldehyde, dehydrated with ethanol, infiltrated with LR White and polymerized at 50 °C. Final samples were sectioned with a diamond ultratome and placed on nickel grids. Grids were blocked with 50 mM Tris-HCl (pH 7.4), 0.5 M NaCl, 0.05% v/v Tween 20 with 3% w/v bovine serum albumin, then incubated with primary antibody[14] (1:10,000) followed by 10 nm gold-conjugated anti-rabbit IgG (1:50). Samples were stained with 2% uranyl acetate; imaging and energy dispersive X-ray analysis was performed using a FEI Technai 20 transmission electron microscope (Center for Advanced Ultrastructural Research, University of Georgia).

**IssA aggregate protein analysis.** The IssA nanoparticle aggregate was immunoprecipitated from P. furiosus cells grown continuously with elemental sulfur[12]. Cells were lysed[40], DNA was sheared using a 21 gauge needle, and protein concentration was brought to 2.5 mg ml$^{-1}$. Cell lysates were pre-cleared with rabbit IgG and then immunoprecipitated with purified IssA antibody cross-linked to Protein A magnetic beads according to the manufacturer's protocol. The immunoprecipitated sample was digested with 10 ng μl$^{-1}$ trypsin at 37 °C overnight and spotted onto a MALDI Anchor plate according to the manufacturer's protocol using NuTip C-18 tips with α-cyano-4-hydroxycinnamic acid matrix. The resulting peptide masses were analysed using MASCOT software searching a Pyrococcus furiosus-specific database and allowing for 2 missed cleavages and a mass difference of ±0.4 Da.

**Native protein purification.** IssA was purified from Pyrococcus furiosus cells grown continuously with $S^0$ (ref. 12). Cells were lysed[40] and centrifuged at 100,000g for 1 h. The pellet was washed first in 50 mM Tris-HCl (pH 8.0), 2 mM dithionite and 2 mM DTT (buffer A) containing 1% w/v sodium dodecyl sulfate, then twice in buffer A and applied to a caesium chloride gradient (density = 1.4 g ml$^{-1}$, 260,000 g, 9.5 h) to remove precipitated material from the media. Fractions containing IssA were pooled and dialyzed against 4l buffer A and concentrated using a Centricon centrifugal concentrator with a 10 kDa cut-off (Millipore). Protein concentration was estimated using the bicinchoninic acid method at 60 °C (ref. 41), following precipitation with 20% w/v trichloroacetic acid. Gel electrophoresis and western blot analyses were performed as described[14].

**Elemental and chemical analyses.** An Agilent 7500c ICP-MS was used to quantify iron and other metals. Sample processing, instrument settings and analysis of mass spectrometry data have been previously described[42]. Colorimetric assays were used to measure iron[43], sulfide[12] and sulfane sulfur[44] in purified protein.

**Negative stain electron microscopy.** Purified IssA exists as a suspension that settles in ~1 h. TEM was conducted on purified IssA as well as on solubilized samples. Solubilization was achieved by adding 20 μl denaturing buffer (6 M guanidinium chloride, 50 mM Tris, 1 mM EDTA, 50 mM DTT) to 20 μl IssA sample. This mixture was shaken at room temperature for 4 h after which the sample was fully dissolved. We diluted the solution six fold with deionized water (dH$_2$O) and applied 3 μl to a glow discharged carbon-coated TEM grid. The grid was washed with 3 μl dH$_2$O and stained with 3 μl of 2% uranyl acetate. Microscopy was conducted on a JEOL 2010 F TEM operated at 200 kV high tension and 50 kX magnification. 100 electron micrographs were recorded in a Gatan Ultrascan 4 K by 4 K CCD camera. We automatically selected 30,000 raw particles and performed 2D image classification in EMAN 2 (ref. 45). Composition of the complexes observed in solubilized IssA was estimated using the volume of a 20 nm sphere, protein density of 1.37 g ml$^{-1}$, FeS density of 4.28 g ml$^{-1}$, a ratio of 38 Fe: 76S: 1 Zn: 1 protein, and the assumption that all volume is occupied by FeS or protein.

**XAS data acquisition and analysis.** Details of iron and sulfur K-edge data collection are described in Supplementary Methods, and the effects of radiation exposure on the sulfur K-edge signal from IssA are shown in Supplementary Fig. 8. The EXAFS oscillations $\chi(k)$ were quantitatively analysed as previously described[46] by curve-fitting using the EXAFSPAK suite of computer programs[47] using *ab initio* theoretical phase and amplitude functions calculated using the programme FEFF version 8.25 (ref. 48). No smoothing, filtering or related operations were performed on the data.

**EPR analysis.** 100 mg purified IssA was loaded into a quartz EPR tube and centrifuged for 30 min. at 1,000$g$. The supernatant was removed and the X-band (~9.6 GHz) EPR spectrum was obtained using a Bruker ESP-300E EPR spectrometer equipped with an ER-4116 dual-mode cavity and an Oxford Instruments ESR-9 flow cryostat.

***In vitro* IssA-mediated reconstitution of apo-Fd from *P. furiosus*.** Apo-Fd from *P. furiosus* was prepared using the method of Moulis and Meyer[49], except that a 2-h incubation with 8% (w/v) TCA at room temperature was required to fully bleach and precipitate the protein. After centrifugation, the apo-protein pellet was dissolved in 200 mM PIPES buffer, pH 6.8, with 1 M NaCl and 1 M KCl, under anaerobic conditions. Apo-Fd (0.25 mM) was incubated with IssA (0.052 mM in monomer, that is, ~8:1 ratio of Fe:Fd) in the same buffer for 24 h at room temperature in the presence of 10 mM DTT, 10 mM sodium dithionite or 10 mM TCEP. The experiment was repeated using the same protocol except that apo-Fd (0.58 mM) was incubated with IssA (0.067 mM in monomer, that is, ~4.4:1 ratio of Fe:Fd) at 80 °C for 1 h. In all cases the resultant Fd was repurified by centrifugation to remove unreacted IssA, buffer exchanged into 50 mM Tris-HCl buffer, pH 7.8, loaded onto a HiTrap Q column and removed as a single band with a 0-1 M NaCl gradient. After desalting and concentrating by Amicon ultrafiltration with a 3 kDa membrane, purity was assessed to be >95% based on gel electrophoresis and the protein concentration was determined using the Bradford assay. [4Fe-4S] cluster content and integrity were assessed compared to the holo-Fd using ultraviolet–visible absorption and CD spectra with ε and Δε values based on protein concentrations.

**Expression and purification of apo-IssA using *E. coli*.** Details of design, expression and purification of the apo-IssA construct are described in Supplementary Methods.

**Apo-IssA reconstitution.** IssA reconstitution was carried out by adding ferrous ammonium sulfate (10 mM) and sodium sulfide (10 mM) in buffer A to apo-IssA (0.25 mM) and incubating with shaking for 1 h at 80 °C. Excess iron and sulfide were removed by buffer exchange using a Centricon concentrator (Millipore) with a 10 kDa cut-off.

**Phylogenetic analysis and structural model.** Details of IPR003731 sequence selection and alignment, phylogenetic tree generation and refinement, construction of the structural model of IssA, and related analyses are described in Supplementary Methods.

**Data availability.** Additional data that support the findings of this study are available from the corresponding author upon reasonable request.

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

## Acknowledgements

This research was supported by a grant from the Division of Chemical Sciences, Geosciences and Biosciences, Office of Basic Energy Sciences of the U.S. Department of Energy (DOE; DE-FG05-95ER20175 to M.W.W.A.) We thank Robert M. Glaeser for providing negative stain electron micrographs and John P. Shields for assistance with whole-cell electron microscopy. Bioinformatic analysis was conducted using the Georgia Advanced Computing Resource Center, a partnership between the University of Georgia's Office of the Vice President for Research and Office of the Vice President for Information Technology. Work at the University of Saskatchewan was supported by the CRC (G.N.G.), NSERC Canada and CIHR (J.J.H.C is a CIHR-THRUST Associate, M.J.H. held a CIHR PDF award and is a CIHR-THRUST fellow). SSRL is supported by the U.S. DOE and NIH. EPR and IssA-mediated apo-Fd reconstitution studies were supported by a grant from the National Institutes of Health (GM62524 to M.K.J.)

## Author contributions

B.J.V., S.M.C., J.F.H., D.-W.L. and M.W.W.A. designed the experiments. B.J.V., S.M.C., J.F.H. and D.-W.L. purified IssA and conducted biochemical analyses. C.-H.W. prepared the recombinant form of IssA. B.J.V. and S.M.C. analysed IssA by TEM *in vivo*. J.S. and H.L. analysed IssA by TEM *in vitro*. J.J.H.C., M.J.H. and G.N.G. conducted XAS measurements and data analysis. B.J.V. and M.K.J. conducted EPR analysis. B.J.V. and F.L.P. conducted bioinformatic analyses. M.K.J. designed and S.M. conducted the IssA-mediated Fd reconstitution experiments. B.J.V. wrote the manuscript in collaboration with S.M.C., M.K.J., G.N.G. and M.W.W.A.

## Additional information

**Competing interests:** The authors declare no competing financial interests.

