## [Peer Review File · Nature Communications]

Reviewer #1 (Remarks to the Author):

This paper reports on a new multi-subunit protein from an hyperthermophilic archaea called IssA and presumably involved in Fe and S storage. The protein has been well characterized by a variety of microscopic techniques since it behaves as particles/aggregates of the same polypeptide chain. It has been purified and XAS data are consistent with the presence of thioferrate polymers within the particles. While this a very interesting and novel observation as it provides a novel insight into the challenging question of Fe storage and delivery to iron sulfur proteins, there are several serious weaknesses which need to be addressed before publication.

In particular the main concern deals with the function of this protein. There is too much speculation in this paper whereas a number of relatively easy experiments should have been included. First, it would be important to find out physiologically relevant conditions (reduction/oxidation?, chelation?, oxidative stress? etc..) for in vitro release of Fe and/or S from IssA, coupled to careful analysis of the products (clusters?). More specifically, using such conditions and using an acceptor protein (a scaffold of the FeS assembly machinery such as IscU or SufBCD or an FeS protein in the apo form), it is also important to see whether this released Fe and S can generate a defined cluster in the acceptor protein. Second, one would like to see, at least in vitro, how an apoIssA, if one can prepare it (see above), can assemble a thioferrate material from Fe and S salts. In the absence of such minimal functional studies the data make an original report of a new protein with an unprecedented biological metal-based polymer but with an unidentified function. This explains why the discussion is so speculative.

Along the same line, the proposed structural model is really too much speculative and should be removed.

Minor concern:

- Please S analysis of the purified protein is needed. If I have read correctly only Fe and Zn (and some other metals) were quantitated.

Reviewer #2 (Remarks to the Author):

Reviewer comments for NCOMMS-16-27617

The manuscript by Vaccaro and coworkers discusses the potential function and structure of a novel iron-sulfur storage protein IssA found in the hyperthermophilic archaeon *Pryococcus furiosus*. Detailed characterization of the IssA protein leads the authors to the conclusion that this protein is storing Fe and S in the form of thioferrate bound to the long C-terminal region of the protein and that the protein forms large homo-oligomers leading to the formation of 20 nm particles that in turn form larger (up to 300 nm) aggregates. The storage of FeS equivalents in the form of thioferrate makes them readily available for the subsequent biosynthesis of different FeS clusters required in many Fe enzymes. The IssA system described in this manuscript is an alternative iron storage system for anaerobic organisms as it does not depend on oxygen to oxidize ferrous iron to produce the ferric form, that is stored in ferritin. The work provides important insights into the Fe cycling in anaerobic organisms and is of high interest to the general readership of Nature Communications.

I recommend some small revisions before publication. These include the following:

Regarding the structural characterization and model:

Regarding the large (300 nm) particles and the formation of 20 nm particles:

It would maybe be useful to use an additional method to characterize the size of the IssA particles if they are eluting at the exclusion limit of the gel filtration columns used. One possible method would be dynamic light scattering. Also it would be interesting if the the authors could discuss the possible oligomerization mechanism in more detail. Are there any indications from the peptide sequence what parts of the structure could be involved in the oligomerization? Is there any packing model how the elongated strands of protein/thioferrate could be arranged in such a particle?

Further points regarding the structural interpretation/model:

Page 9, line 190ff and Fig. 5: There is a relevant subgroup of proteins included in the IssA clade in Fig. 5 that show acidic pI in the range of 5-6. Are these thought to be part of the the IssA family as well? Please comment.

More details about conservation of the cationic residues in the tail region (Page 11,222-228) would be useful. What is the situation for other members of the IssA protein clade?

Page 16, line 346: Please comment on what this homologous protein is. Is there any connection to the IssA family?

Regarding the XAS data:

Page 8. line 162 and Fig. 3: suggestion: show also XAS for a thioferrate as a relevant model for the proposed cluster in IssA.

Figure 3A: suggestion: add a schematic of the different Fe centers for the three reference compounds b,c,d in the figure as an inset.

Figure 3B: Is the signal level of the 5.4 Å peak (page 7, line 152) high enough above the noise for a clear interpretation? What about the similar size peak at 3.7 Å (Fig 3B)? What is the cause of the peak at 3.3 Å in the S EXAFS? The fit seems not to reproduce that peak. Is that only noise? Please explain in more detail.

- page 20, line 434: Was there any indication of radiation damage to the sample? Adding 16 scans of 30 minutes on a single sample spot at 2.5 keV could lead to significant radiation damage to the sample. Please explain in more detail how damage was checked or if the sample was changed after each scan.

Regarding the EPR data:

- Page 8, line 169 and fig. S5: The signal looks not centered at $g=2$ but rather at slightly higher g . Please correct accordingly.
- Page 8, line 174: Shouldn't vanadium show up in the elemental analysis if it is that clearly visible in the EPR (Table S1, Fig. S2)? Please comment/explain.

Regarding the functional interpretation:

Did the authors characterize the phenotype of a Δ IssA mutant? The observation of a knock out phenotype would provide additional support for the functional role of IssA discussed in this manuscript and such data should be included if possible.

small corrections:

- page 2, line 31, change "Growth on the organism" to "Growth of the organism"
- Page 16, line 328: delete the 2nd "computed" in this sentence
- page 18, line 376: fix reference formatting in "S012"
- page 20, line 421, what was the temperature at the sample with the He cryostream?

With best regards,

Jan Kern

Reviewer #3 (Remarks to the Author):

The manuscript shows the isolation and characterization of a novel thioferrate storing protein in the hyperthermophilic archaeon *Pyrococcus furiosus*. This protein forms giant particles in the cytoplasm of the organism when it is grown with elemental sulfur. The manuscript rigourously describes the isolation and characterization of this protein. This work appears to be done very well and the results are genuinely surprising and exciting, although I am not an expert on most of these methods. I will therefore restrict my critical comments to the phylogenetic analysis.

The phylogenetic analysis is conducted by first aligning the ~5000 homologs of the *IssA* protein with Clustal Omega and producing a phylogenetic tree from the alignment using Clustal W. Although the tree that is produced is probably not terrible, this is definitely not the state-of-the-art in phylogenetic analysis. The authors should be using actual phylogenetic software intended for this purpose -- the ClustalW tree making capabilities are really to generate guide trees for alignment, not for rigorous analysis. In any case, the authors should take their alignment and analyze it with proper phylogenetic program like RAxML or IQTREE. They should use model selection techniques implemented in these software tools to select the most appropriate substitution models for analysis (including models that allow for gamma rates-across-sites etc). They should also conduct bootstrap analysis. Any phylogenetic points they make in the manuscript should make reference to branches on these trees and the bootstrap support that they receive (i.e. the '*IssA*' clade and other clades identified). Such analyses should be fairly straightforward to conduct and are far more rigorous than those described. The authors should also consider 'trimming' their alignments using software like ZORRO, trimAL or BMGE. The trimming removes regions of inherently poor alignment and should remove statistical noise.

Responses to Reviewers' Comments

Reviewer #1

This paper reports on a new multi-subunit protein from an hyperthermophilic archaea called IssA and presumably involved in Fe and S storage. The protein has been well characterized by a variety of microscopic techniques since it behaves as particles/aggregates of the same polypeptide chain. It has been purified and XAS data are consistent with the presence of thioferrate polymers within the particles. While this a very interesting and novel observation as it provides a novel insight into the challenging question of Fe storage and delivery to iron sulfur proteins, there are several serious weaknesses which need to be addressed before publication.

In particular the main concern deals with the function of this protein. There is too much speculation in this paper whereas a number of relatively easy experiments should have been included. First, it would be important to find out physiologically relevant conditions (reduction/oxidation?, chelation?, oxidative stress? etc..) for *in vitro* release of Fe and/or S from IssA, coupled to careful analysis of the products (clusters?). More specifically, using such conditions and using an acceptor protein (a scaffold of the FeS assembly machinery such as IscU or SufBCD or an FeS protein in the apo form), it is also important to see whether this released Fe and S can generate a defined cluster in the acceptor protein.

Response: The reviewer makes an excellent point as to whether IssA-thioferrate complex can be used to assemble Fe-S clusters in *P. furiosus*. We have now shown experimentally that *in vitro* this is indeed the case. We demonstrated ~50% IssA-mediated reconstitution of the [4Fe-4S]²⁺ cluster in the apo-form of the ferredoxin from *P. furiosus* under anaerobic conditions. The reaction occurred over 24 hours at room temperature and over 1 hour at the more physiological temperature of 80 °C. We choose the ferredoxin as *P. furiosus* almost exclusively contains [4Fe-4S] cluster-containing Fe-S proteins and the [4Fe-4S]-containing ferredoxin is one of the most abundant Fe-S protein in the organism. Importantly, the cluster reconstitution using apo-ferredoxin and IssA occurred in the presence of disulfide-reducing agent DTT but not in the presence of the reductant dithionite or the alternative disulfide-reducing reagent TCEP. We therefore postulate that DTT is replacing an as yet unknown cellular thiol present *in vivo* that allows the thioferrate in IssA to provide Fe and S for [4Fe-4S] cluster synthesis. These results and their implications have now been incorporated into a new section in the results section, along with a new figure (Figure 5). We also point out that *P. furiosus* does not contain an IscU-type protein and the only potential Fe-S cluster assembly proteins are the SufCBD complex. However, the putative SufB scaffold protein has only two cysteines (compared to 13 in *E. coli* SufB), which raises questions about a potential role for SufB as a scaffold protein in *P. furiosus*. Hence, the scaffolding hypothesis that constitutes the current paradigm for Fe-S cluster assembly in other organisms may not apply in *P. furiosus* under iron- and sulfide-replete conditions due to the presence of IssA.

Second, one would like to see, at least *in vitro*, how an apoIssA, if one can prepare it (see above), can assemble a thioferrate material from Fe and S salts. In the absence of such minimal functional studies the data make an original report of a new protein with an unprecedented biological metal-based polymer but with an unidentified function. This explains why the discussion is so speculative.

Response: To determine whether the globular IPR003731 domain of IssA would bind Fe and S, analogous to its Nif relatives, and oligomerize, the His-tagged recombinant form was produced *in*

E. coli (apo-IssA). Mixing soluble apo-IssA with Fe and S salts resulted in formation of a high molecular weight Fe-S complexes (up to 900 kDa in size). These results and the implications are now described in the text.

Along the same line, the proposed structural model is really too much speculative and should be removed.

Response: We agree with the reviewer and have deleted the structural model (original Figure 6). The threaded model of the IPR003731 domain of *P. furiosus* IssA is now shown with a C-terminal extension that is clearly not intended to convey any structural prediction (see Figure 6). We have retained some discussion on the unique structural features that are present in IssA, based on structures of homologs of the N-terminal domain and the need to stabilize the negatively-charged thioferrate chain.

Minor concern:

- Please S analysis of the purified protein is needed. If I have read correctly only Fe and Zn (and some other metals) were quantitated.

Response: The reviewer is correct. Acid-labile sulfide and sulfane sulfur assays have now been carried out and the results are now included. Interestingly, the measured sulfide and sulfur values give additional support to the presence of a thioferrate-type structure in IssA (based on the reactivity of thioferrates under assay conditions) and this is now discussed.

Reviewer #2

The manuscript by Vaccaro and coworkers discusses the potential function and structure of a novel iron-sulfur storage protein IssA found in the hyperthermophilic archaeon *Pryococcus furiosus*. Detailed characterization of the IssA protein leads the authors to the conclusion that this protein is storing Fe and S in the form of thioferrate bound to the long C-terminal region of the protein and that the protein forms large homo-oligomers leading to the formation of 20 nm particles that in turn form larger (up to 300 nm) aggregates. The storage of FeS equivalents in the form of thioferrate makes them readily available for the subsequent biosynthesis of different FeS clusters required in many Fe enzymes. The IssA system described in this manuscript is an alternative iron storage system for anaerobic organisms as it does not depend on oxygen to oxidize ferrous iron to produce the ferric form, that is stored in ferritin. The work provides important insights into the Fe cycling in anaerobic organisms and is of high interest to the general readership of *Nature Communications*. I recommend some small revisions before publication.

These include the following:

Regarding the large (300 nm) particles and the formation of 20 nm particles: It would maybe be useful to use an additional method to characterize the size of the IssA particles if they are eluting at the exclusion limit of the gel filtration columns used. One possible method would be dynamic light scattering.

Response: We have now attempted dynamic light scattering analysis of IssA. However, IssA was not a good candidate for this type of analysis due to its high density (it exists as a suspension), high extinction coefficient (samples appear black) and non-uniform size. A description of these results has been included in the manuscript.

Also it would be interesting if the the authors could discuss the possible oligomerization mechanism in more detail. Are there any indications from the peptide sequence what parts of the structure could be involved in the oligomerization? Is there any packing model how the elongated strands of protein/thioferrate could be arranged in such a particle?

Response: As described in the response to Reviewer 1, we have shown using the recombinant form expressed in *E. coli*, that the globular IPR003731 domain of IssA is able to oligomerize in the presence of Fe and S. These results have been included in the manuscript.

Further points regarding the structural interpretation/model: Page 9, line 190ff and Fig. 5: There is a relevant subgroup of proteins included in the IssA clade in Fig. 5 that show acidic pI in the range of 5-6. Are these thought to be part of the the IssA family as well? Please comment.

Response: Comments specifically addressing lower pI members of the IssA clade have been added based on the new phylogenetic analyses (suggested by Reviewer 3).

More details about conservation of the cationic residues in the tail region (Page 11,222-228) would be useful. What is the situation for other members of the IssA protein clade?

Response: Additional detail has been included in both the results and discussion sections, based on the new phylogenetic analyses (suggested by Reviewer 3).

Page 16, line 346: Please comment on what this homologous protein is. Is there any connection to the IssA family?

Response: The homologous protein is a closely-related member of the IssA family (containing IPR003731). This is now specifically indicate in the new Figure 6.

Regarding the XAS data: Page 8. line 162 and Fig. 3: suggestion: show also XAS for a thioferrate as a relevant model for the proposed cluster in IssA.

Response: Sulfur K-edge XAS of thioferrate has been reported by others and we now discuss this in the Discussion section.

Figure 3A: suggestion: add a schematic of the different Fe centers for the three reference compounds b,c,d in the figure as an inset.

Response: This is a good suggestion and it has been added to a revised version of the Figure.

Figure 3B: Is the signal level of the 5.4 Å peak (page 7, line 152) high enough above the noise for a clear interpretation? What about the similar size peak at 3.7Å (Fig 3B)? What is the cause of the peak at 3.3 Å in the S EXAFS? The fit seems not to reproduce that peak. Is that only noise? Please explain in more detail.

Response: The reviewer is correct as the 5.4 Å transform peak is close to the noise level, and we have inserted additional text on page 8 to address these concerns. Specifically, “This 5.4 Å interaction is close to the limit of the noise, being about three times the transform peak height of the noise as estimated from higher *R* values, using data to 18 Å⁻¹. However, it shows behavior characteristic of real EXAFS, rather than a noise peak in the Fourier transform, so that the feature persists irrespective of the *k*-ranges, and moreover fits to a very similar Fe····Fe distance with different *k*-ranges. The use of multiple scattering EXAFS reproduces many weaker features in the EXAFS (Fig. 3b) that also appear to be above the noise level (see supplementary Fig. S5). Other weak interactions in the EXAFS data, such as the 3.3 Å feature in the S K-edge EXAFS, do not behave in this manner and these are likely due to noise (not illustrated).”

- page 20, line 434: Was there any indication of radiation damage to the sample? Adding 16 scans of 30 minutes on a single sample spot at 2.5 keV could lead to significant radiation damage to the sample. Please explain in more detail how damage was checked or if the sample was changed after each scan.

Response: This is an excellent point. We averaged data was combined from three different samples (5 sweeps, 5 sweeps and 6 sweeps) and the near-edge portion of the spectrum was monitored for changes indicating radiation damage. This is now elaborated upon in the methods

section on page 21 of the revised manuscript and an additional reference is cited. A supplemental Figure (S7) has been added to show that only minor changes arise from X-ray induced radiation damage with increasing exposure time.

Regarding the EPR data - Page 8. line 169 and fig. S5: The signal looks not centered at $g=2$ but rather at slightly higher g . Please correct accordingly.

Response: The reviewer is correct, the resonance is centered just above $g = 2$. This has been changed in the text and the supplementary figure, now S6.

- Page 8, line 174: Shouldn't vanadium show up in the elemental analysis if it is that clearly visible in the EPR (Table S1, Fig. S2)? Please comment/explain.

Response: The reviewer is correct. ICP-MS analyses indicated up to 0.004 atoms of V per IssA. This concentration is too small to explain the intensity of the weak $S = 1/2$ EPR signal centered at $g = 2.0$. We have removed this tentative assignment in the text and the supplementary figure, now S6.

Regarding the functional interpretation: Did the authors characterize the phenotype of a delta-IssA mutant? The observation of a knock out phenotype would provide additional support for the functional role of IssA discussed in this manuscript and such data should be included if possible.

Response: We could not detect a phenotype for the IssA deletion mutant during growth with or without S^0 . Since IssA is only expressed under iron-replete conditions, attempts were made to observe a phenotype during transition to growth on an iron-deficient medium, but this was also unsuccessful.

small corrections:

- page 2, line 31, change "Growth on the organism" to "Growth of the organism"

Response: This error has been corrected.

- Page 16, line 328: delete the 2nd "computed" in this sentence

Response: This error has been corrected.

- page 18, line 376: fix reference formatting in "S012"

Response: This has been clarified.

- page 20, line 421, what was the temperature at the sample with the He cryostream?

Response: We estimate a sample temperature of 20-50 K based on a cryostream temperature of 20 K, and this information has been included in the manuscript.

Reviewer #3

The manuscript shows the isolation and characterization of a novel thioferrate storing protein in the hyperthermophilic archaeon *Pyrococcus furiosus*. This protein forms giant particles in the cytoplasm of the organism when it is grown with elemental sulfur. The manuscript rigourously describes the isolation and characterization of this protein. This work appears to be done very well and the results are genuinely surprising and exciting, although I am not an expert on most of these methods. I will therefore restrict my critical comments to the phylogenetic analysis.

The phylogenetic analysis is conducted by first aligning the ~5000 homologs of the IssA protein with Clustal Omega and producing a phylogenetic tree from the alignment using Clustal W.

Although the tree that is produced is probably not terrible, this is definitely not the state-of-the-art in phylogenetic analysis. The authors should be using actual phylogenetic software intended for this purpose -- the ClustalW tree making capabilities are really to generate guide trees for alignment, not for rigorous analysis. In any case, the authors should take their alignment and analyze it with proper phylogenetic program like RAxML or IQTREE. They should use model selection techniques implemented in these software tools to select the most appropriate substitution models for analysis (including models that allow for gamma rates-across-sites etc). They should also conduct bootstrap analysis. Any phylogenetic points they make in the manuscript should make reference to branches on these trees and the bootstrap support that they receive (i.e. the 'IssA' clade and other clades identified). Such analyses should be fairly straightforward to conduct and are far more rigorous than those described. The authors should also consider 'trimming' their alignments using software like ZORRO, trimAL or BMGE. The trimming removes regions of inherently poor alignment and should remove statistical noise.

Response: We thank the reviewer for these very useful comments. A much more rigorous tree construction has been now conducted using the recommended IQ-Tree (including automated model selection and bootstrapping). We also took advantage of conservative trimming using the trimAl program, and the corresponding sections in the text have been updated accordingly. We have restricted tree display to branches with 70% confidence or higher bootstrap scores and report the bootstrap confidence values for all clades discussed.

Reviewer #1 (Remarks to the Author):

The authors have nicely addressed my concerns including via an extensive addition of experiments and data, which provide some support to suggestions regarding the function of these protein particles.

I recommend publication

Reviewer #2 (Remarks to the Author):

The revised manuscript by Vaccaro et al addressed the points I raised in the previous version and the newly added reconstitution data strengthen the conclusion of the manuscript that IssA can be involved in Fe-cluster assembly.

The manuscript is acceptable for publication in its current form..

I have only a few minor errors that should be addressed in the final version:

Page 9, line 6 from bottom:

"sulfide and sulfane sufur content" should be "sulfide and sulfane sulfur content"

Page 10, line 3:

"At 60 K, the spectra are show weak resonance"

change to

"At 60 K, the spectra show weak resonance"

Page 12, line 3:

"residues forms a tail region" change to "residues form a tail region"

Page 13, line 5 from bottom:

"cationic (7 of 70), aromatic (14 of 70) and glycine (15 of 70). "

should be

"cationic (7 of 70), aromatic (14 of 70) and glycine (15 of 70) residues."

Jan Kern

Reviewer #3 (Remarks to the Author):

I have reviewed the phylogenetic analyses reported and discussed in the revised manuscript and I think they are now up to standard. The authors are appropriate in reporting support values in the text for the protein families they describe and their conclusions from the phylogenies are well justified.

REVIEWERS' COMMENTS:

Reviewer #1 (Remarks to the Author):

The authors have nicely addressed my concerns including via an extensive addition of experiments and data, which provide some support to suggestions regarding the function of these protein particles. I recommend publication

Response: We thank the reviewer for their very helpful comments throughout the review process.

Reviewer #2 (Remarks to the Author):

The revised manuscript by Vaccaro et al addressed the points I raised in the previous version and the newly added reconstitution data strengthen the conclusion of the manuscript that IssA can be involved in Fe-cluster assembly.

Response: We thank the reviewer for their very helpful comments throughout the review process.

The manuscript is acceptable for publication in its current form. I have only a few minor errors that should be addressed in the final version:

Page 9, line 6 from bottom:

"sulfide and sulfane sufur content" should be "sulfide and sulfane sulfur content"

Response: this has been corrected.

Page 10, line 3:

"At 60 K, the spectra are show weak resonance"
change to

"At 60 K, the spectra show weak resonance"

Response: this has been corrected.

Page 12, line 3:

"residues forms a tail region" change to "residues form a tail region"

Response: this has been corrected.

Page 13, line 5 from bottom:

"cationic (7 of 70), aromatic (14 of 70) and glycine (15 of 70). "
should be

"cationic (7 of 70), aromatic (14 of 70) and glycine (15 of 70) residues."

Response: this has been corrected.

Reviewer #3 (Remarks to the Author):

I have reviewed the phylogenetic analyses reported and discussed in the revised manuscript and I think they are now up to standard. The authors are appropriate in reporting support values in the text for the protein families they describe and their conclusions from the phylogenies are well justified.

Response: We thank the reviewer for their very helpful comments throughout the review process.